# Probabilistic Graphical Model for Robust Graph Neural Networks against Noisy Labels

## Abstract

While robust graph neural networks (GNNs) have been widely studied for graph perturbation and attack, those for label noise have received significantly less attention. Most existing methods heavily rely on the label smoothness assumption to correct noisy labels, which adversely affects their performance on heterophilous graphs. Further, they generally perform poorly in high noise-rate scenarios. To address these problems, in this paper, we propose a novel probabilistic graphical model based framework PRGNN. Given a noisy label set and a clean label set, our goal is to maximize the likelihood of labels in the clean set. We first present PRGNN-v1, which generates clean labels based on graphs only in the Bayesian network. To further leverage the information of clean labels in the noisy label set, we put forward PRGNN-v2, which incorporates the noisy label set into the Bayesian network to generate clean labels. The generative process can then be used to predict labels for unlabeled nodes. We conduct extensive experiments to show the robustness of PRGNN on varying noise types and rates, and also on graphs with different heterophilies. In particular, we show that PRGNN can lead to inspiring performance in high noise-rate situations. The implemented code is available at `https://github.com/PRGNN/PRGNN`.

## 1 Introduction

Graph Neural Networks (GNNs) have been widely applied in a variety of fields, such as social network analysis Hamilton et al. (2017), drug discovery Li et al. (2021a), financial risk control Wang et al. (2019), and recommender systems Wu et al. (2022). However, real-world graph data often contain noisy labels, which are generally derived from inadvertent errors in manual labeling on crowdsourcing platforms or incomplete and inaccurate node features corresponding to labels. These noisy labels have been shown to degenerate the performance of GNNs Zhang et al. (2021); Patrini et al. (2017) and further reduce the reliability of downstream graph analytic tasks. Therefore, tackling label noise for GNNs is a critical problem to be addressed.

Recently, label noise has been widely studied in the field of Computer Vision (CV) Cheng et al. (2020); Yi et al. (2022); Han et al. (2018); Li et al. (2020); Shu et al. (2019), which aims to derive robust neural network models. Despite the success, most existing methods cannot be directly applied to graph-structured data due to the inherent non-Euclidean characteristics and structural connectivity of graphs. Although some methods specifically designed for graphs have shown promising results Nt et al. (2019); Li et al. (2021b); Du et al. (2021); Dai et al. (2021); Xia et al. (2021a); Qian et al. (2023), they still suffer from two main limitations. First, most existing approaches heavily rely on label smoothness to correct noisy labels, which assumes that neighboring nodes in a graph tend to have the same label. This assumption is typically used to express local continuity in homophilous graphs and does not hold in heterophilous graphs. When applied in graphs with heterophily, the performance of these methods could be significantly degraded. Second, while probabilistic graphical models have been successively used to handle label noise in CV, there remains a gap in applying them for GNNs against noisy labels. It is well known that probabilistic graphical model and Bayesian framework can model uncertainty and are thus less sensitive to data noise. Therefore, there arises a question: *Can we develop a probabilistic graphical model for robust GNNs against noisy labels?*

In this paper, we study robust GNNs from a Bayesian perspective. Since it is generally easy to obtain an additional small set of clean labels at low cost, we consider a problem scenario that includes both a noisy training set and a clean one of much smaller size. We propose a novel framework based on **P**robabilistic graphical model for **R**obust **GNN**s against noisy labels, namely, *PRGNN*. We emphasize that PRGNN does not assume label smoothness, and can be applied in both graphs with homophily and heterophily. Given a noisy label set $Y_N$ and a much smaller clean label set $Y_C$ in a graph $G$, our goal is to maximize the likelihood of clean labels in $Y_C$. To reduce the adverse effect from noise in $Y_N$, PRGNN-v1 (version 1) maximizes $P(Y_C|G)$, which assumes the conditional dependence of $Y_C$ on $G$ only in the Bayesian network. Specifically, PRGNN-v1 first introduces a hidden variable $\bar{Y}$ that expresses noisy labels for nodes, and then generates clean labels $Y_C$ based on both $G$ and $\bar{Y}$. Note that PRGNN-v1 implicitly restricts the closeness between $\bar{Y}$ and $Y_N$ to take advantage of informative clean labels in $Y_N$. To better use $Y_N$, we further present PRGNN-v2 (version 2), which assumes the conditional dependence of $Y_C$ on both $G$ and $Y_N$, and maximizes $P(Y_C|G, Y_N)$. The simultaneous usage of $G$ and $Y_N$ can lead to less noisy $\bar{Y}$ and further improves the accuracy of $Y_C$ generation. To maximize the likelihood, we employ the variational inference framework and derive ELBOs as objectives in both PRGNN-v1 and PRGNN-v2. In particular, we use three independent GNNs to implement the encoder that generates $\bar{Y}$, the decoder that generates $Y_C$, and the prior knowledge of $\bar{Y}$, respectively. Since node raw features and labels in $Y_C$ or $Y_N$ could be in different semantic space, directly concatenating features with one-hot encoded labels as inputs of GNNs could result in undesired results. To solve the issue, we first perform GNNs on raw features to generate node embeddings, based on which label prototype vectors are then calculated. In this way, node features and labels can be inherently mapped into the same low-dimensional space. After that, we fuse node embeddings and label prototype vectors to generate both $\bar{Y}$ and $Y_C$. During the optimization, we highlight clean labels while attenuating the adverse effect of noisy labels in $Y_N$. Finally, we summarize our main contributions in this paper as:

- We propose PRGNN, which is the first probabilistic graphical model based framework for robust GNNs against noisy labels, to our best knowledge.
- We disregard the label smoothness assumption for noise correction, which leads to the wide applicability of PRGNN in both homophilous and heterophilous graphs.
- We extensively demonstrate the effectiveness of PRGNN on different benchmark datasets, GNN architectures, and various noise types and rates. In particular, we show that PRGNN can lead to inspiring performance in high noise-rate situations.

## 2 RELATED WORK

### 2.1 DEEP NEURAL NETWORKS WITH NOISY LABELS

Learning with noisy labels has been widely studied in CV. From Song et al. (2022), most existing methods can be summarized in the following five categories: Robust architecture Cheng et al. (2020); Yao et al. (2018); Robust regularization Yi et al. (2022); Xia et al. (2021b); Wei et al. (2021); Robust loss function Ma et al. (2020); Zhang & Sabuncu (2018); Loss adjustment Huang et al. (2020); Wang et al. (2020b); Sample selection Han et al. (2018); Yu et al. (2019b); Li et al. (2020); Wei et al. (2020). However, the aforementioned approaches are dedicated to identically distributed (i.i.d) data, which may not be directly applicable to GNNs for handing noisy labels because the noisy information can propagate via message passing of GNNs.

### 2.2 ROBUST GRAPH NEURAL NETWORKS

In recent years, GNN has gained significant attention due to its broad range of applications in downstream tasks, such as node classification Oono & Suzuki (2019), link prediction Baek et al. (2020), graph classification Errica et al. (2019), and feature reconstruction Hou et al. (2022). Generally, existing robust GNN methods can be mainly divided into two categories: one that deals with perturbed graph structures and node features Zhu et al. (2021); Zhang & Zitnik (2020); Yu et al. (2021); Wang et al. (2020a), while the other that handles noisy labels. In this paper, we focus on solving the problem of the latter and only few works have been proposed. For example, D-GNN Nt et al. (2019) applies the backward loss correction to reduce the effects of noisy labels. UnionNET Li et al. (2021b) performs label aggregation to estimate node-level class probability distributions, which are

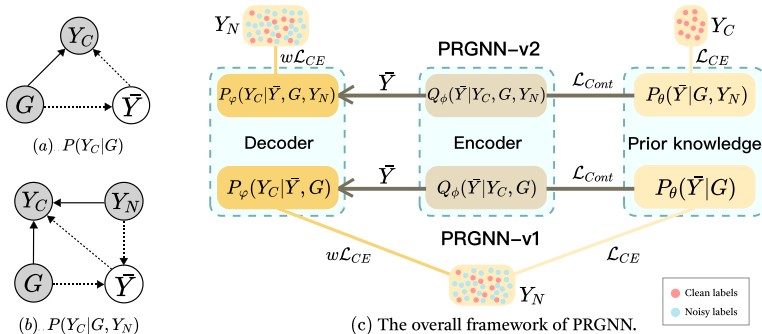

Figure 1: Bayesian networks of (a) $P(Y_C|G)$ and (b) $P(Y_C|G, Y_N)$. Here, $G$ is the input graph data, $Y_C$ is the clean label set, $Y_N$ is the noisy label set, and $\bar{Y}$ is the hidden variable. Arrows with solid lines and dashed lines denote generative process and inference process, respectively.

used to guide sample reweighting and label correction. PIGNN Du et al. (2021) leverages the PI (Pairwise Interactions) between nodes to explicitly adjust the similarity of those node embeddings during training. To alleviate the negative effect of the collected sub-optimal PI labels, PIGNN further introduces a new uncertainty-aware training approach and reweights the PI learning objective by its prediction confidence. NRGNN Dai et al. (2021) connects labeled nodes with high similarity and unlabeled nodes, constructing a new adjacency matrix to train more accurate pseudo-labels. LPM Xia et al. (2021a) computes pseudo labels from the neighboring labels for each node in the training set using Label Propagation (LP) and utilizes meta learning to learn a proper aggregation of the original and pseudo labels as the final label. RTGNN Qian et al. (2023) is based on the hypothesis that clean labels and incorrect labels in the training set are given, which is generally difficult to satisfy in reality.

Despite their success, we observe that most of them heavily rely on the label smoothness assumption, so that they cannot be applied to heterophilous graphs. In addition, most of them perform poorly in high noise-rate. Different from these methods, our proposed method PRGNN can achieve superior performance under different noise types and rates on various datasets.

## 3 PRELIMINARY

We denote a graph as $G = (V, E)$, where $V = \{v_i\}_{i=1}^n$ is a set of nodes and $E \subseteq V \times V$ is a set of edges. Let $A$ be the adjacency matrix of $G$ such that $A_{ij}$ represents the weight of edge $e_{ij}$ between nodes $v_i$ and $v_j$. For simplicity, we set $A_{ij} = 1$ if $e_{ij} \in E$; 0, otherwise. Nodes in the graph are usually associated with features and we denote $X$ as the feature matrix, where the $i$-th row $x_i$ indicates the feature vector of node $v_i$.

**Definition 1** *Given a graph $G$ that contains a small clean training set $\mathcal{T}_C$ with labels $Y_C$ and a noisy training set $\mathcal{T}_N$ with labels $Y_N$, where $|\mathcal{T}_C| \ll |\mathcal{T}_N|$, our task is to learn a robust GNN $f(\cdot)$ that can predict the labels $Y_U$ of unlabeled nodes, i.e.,*

$$f(\mathcal{G}, \mathcal{T}_C, \mathcal{T}_N) \to Y_U. \tag{1}$$

## 4 METHODOLOGY

### 4.1 PRGNN-v1

To predict $Y_U$ for unlabeled nodes, we need to calculate the posterior distribution $P(Y_U|G, Y_C, Y_N)$. Instead of calculating the posterior directly, we propose to maximize the likelihood of $P(Y_C|G)$, which aims to generate the informative clean labels $Y_C$. The generative process can then be used to predict $Y_U$. The Bayesian network for generating $Y_C$ is shown in Figure 1(a) and the generative process is formulated as:

$$P(Y_C|G) = \int_{\bar{Y}} P(Y_C|\bar{Y}, G)P(\bar{Y}|G)dY. \tag{2}$$

Generally, the hidden variable $\bar{Y}$ can be interpreted as node embedding matrix. Since the matrix is then used to predict node labels, we directly denote $\bar{Y}$ as noisy label predictions for all the nodes in the graph. The generative process can be described as follows: we first obtain noisy label predictions $\bar{Y}$ for all the nodes in the graph, and then jointly consider $G$ and $\bar{Y}$ to generate the true clean labels $Y_C$. Since directly optimizing $P(Y_C|G)$ is difficult, we introduce a variational distribution $Q_\phi(\bar{Y}|Y_C, G)$ with parameters $\phi$ and employ variational inference to derive the evidence lower bound (ELBO) as:

$$logP_{\theta,\varphi}(Y_C|G) \geq E_{Q_\phi(\bar{Y}|Y_C,G))}logP_\varphi(Y_C|\bar{Y}, G) - KL(Q_\phi(\bar{Y}|Y_C,G)||P_\theta(\bar{Y}|G)) = \mathcal{L}^1_{ELBO}. \tag{3}$$

Here, $Q_\phi(\bar{Y}|Y_C, G)$ characterizes the encoding (mapping) process, while $P_\varphi(Y_C|\bar{Y}, G)$ represents the decoding (reconstruction) process. Note that $P_\varphi$ can generate predicted labels $\hat{Y}$ for all the nodes more than $\hat{Y}_C$. Further, $P_\theta(\bar{Y}|G)$ captures the prior knowledge. In our experiments, we use three independent GNNs to implement them with learnable parameters $\phi$, $\varphi$ and $\theta$, respectively. However, the above generative process ignores the given noisy labels $Y_N$, while $Y_N$ still contains many clean node labels that are informative. To further employ the useful information from $Y_N$, we first apply a standard multiclass softmax cross-entropy loss $\mathcal{L}_{CE}(P_\theta, Y_N)$ to incorporate $Y_N$ into the prior knowledge $P_\theta(\bar{Y}|G)$. In addition to $Y_C$, for nodes with clean labels in $Y_N$, it is also expected that their reconstructed labels should be close to their ground-truth ones. However, clean node labels are unknown in $Y_N$. To address the problem, we use the similarity between $Y_N$ and $\hat{Y}_N$, denoted as $w \in \mathbb{R}^{|Y_N|}$, to measure the degree to which a node label in $Y_N$ is clean. Intuitively, for a labeled node $v_i$ in the noisy training set, if its reconstructed label $\hat{y}_i$ is similar as its label $y_i \in Y_N$, it is more likely that $y_i$ is a clean label; otherwise not. After that, we adopt a weighted cross-entropy loss $w\mathcal{L}_{CE}(P_\varphi, Y_N)$, which assigns large weights to clean nodes while attenuating the erroneous effects of noisy labels. In addition, to leverage the extra supervision information from massive unlabeled data, inspired by Wan et al. (2021), we add the contrastive loss $\mathcal{L}_{Cont}(Q_\phi, P_\theta)$ to maximize the agreement of predictions of the same node that are generated from $Q_\phi(\bar{Y}|Y_C, G)$ and $P_\theta(\bar{Y}|G)$. Due to the space limitation, we defer details on $\mathcal{L}_{Cont}$ to the Appendix C. Finally, the overall loss function is formulated as:

$$\mathcal{L}_1(\theta,\varphi,\phi) = -\mathcal{L}^1_{ELBO}(\theta,\varphi,\phi) + \lambda_1 w\mathcal{L}_{CE}(P_\varphi, Y_N) + \lambda_2\mathcal{L}_{CE}(P_\theta, Y_N) + \lambda_3\mathcal{L}_{Cont}(Q_\phi, P_\theta) \tag{4}$$

where $\lambda_1$, $\lambda_2$ and $\lambda_3$ are hyper-parameters to balance the losses.

## 4.2 PRGNN-v2

In Section 4.1, PRGNN-v1 leverages $Y_N$ from two aspects. On the one hand, $Y_N$ is considered as the prior knowledge and incorporated into $P_\theta$. On the other hand, for nodes with clean labels in $Y_N$, their predicted labels are enforced to be close to the clean ones. However, $Y_N$ is not directly included in the generative process of $Y_C$ (see Figure 1(a)), and $Y_C$ is only determined by $G$ and the hidden variable $\bar{Y}$. From Equation 4, we see that PRGNN-v1 implicitly restricts the closeness between $\bar{Y}$ and $Y_N$ with regularization terms. In this way, when the number of erroneous labels in $Y_N$ is large, $\bar{Y}$ will be noisy and further degrade the performance of generating $Y_C$. To address the problem, we propose PRGNN-v2, which is a probabilistic graphical model using $Y_N$ to generate $Y_C$ (see Figure 1(b)). The goal of PRGNN-v2 is to maximize $P(Y_C|G, Y_N)$. Similarly, we introduce a variational distribution $Q_\phi(\bar{Y}|Y_C, G, Y_N)$ and derive the ELBO as:

$$logP_{\theta,\varphi,\phi}(Y_C|G, Y_N)$$
$$\geq E_{Q_\phi(\bar{Y}|Y_C,G,Y_N)}logP_\varphi(Y_C|\bar{Y}, G, Y_N) - KL(Q_\phi(\bar{Y}|Y_C,G,Y_N)||P_\theta(\bar{Y}|G, Y_N)) = \mathcal{L}^2_{ELBO}. \tag{5}$$

In our experiments, we also use three independent GNNs to implement $P_\theta$, $Q_\phi$ and $P_\varphi$, respectively. Note that the prior knowledge $P_\theta(\bar{Y}|G, Y_N)$ is a conditional distribution based on $G$ and $Y_N$, so $\bar{Y}$ is easily to be adversely affected by the noise label in $Y_N$. To reduce noise in $\bar{Y}$, we explicitly use cross-entropy loss $\mathcal{L}_{CE}(P_\theta, Y_C)$ to force $\bar{Y}$ to be close to $Y_C$[1]. Similar as Equation 4, we formulate the overall objective by further adding a weighted cross-entropy term and a contrastive loss term:

$$\mathcal{L}_2(\theta,\varphi,\phi) = -\mathcal{L}^2_{ELBO}(\theta,\varphi,\phi) + \lambda_1 w\mathcal{L}_{CE}(P_\varphi, Y_N) + \lambda_2\mathcal{L}_{CE}(P_\theta, Y_C) + \lambda_3\mathcal{L}_{Cont}(Q_\phi, P_\theta). \tag{6}$$

---

[1] We do not explicitly add the term in Equation 4 because $Y_N$ is not used as a condition in $P_\theta$.

Different from PRGNN-v1, PRGNN-v2 explicitly adds $Y_N$ in the Bayesian network to generate $Y_C$. Instead of enforcing the closeness between $\bar{Y}$ and $Y_N$, PRGNN-v2 leverages the power of GNNs to correct noise labels in $Y_N$ and obtain a high-quality $\bar{Y}$, leading to better reconstruction of $Y_C$.

### 4.3 ENCODER

In the encoder, we generate the hidden variable $\bar{Y}$ based on $Q_\phi(\bar{Y}|Y_C, G)$ or $Q_\phi(\bar{Y}|Y_C, G, Y_N)$. A naive solution is to use one-hot encoded embeddings for labels in $Y_C$ and $Y_N$, and concatenate them with raw node features, which are further fed into GNNs to output $\bar{Y}^2$. However, labels and raw node features may correspond to different semantic spaces, which could adversely affect the model performance. To solve the issue, we employ label prototype vectors to ensure that labels and nodes are embedded into the same low-dimensional space. Specifically, we first run a GNN model on $G$ to generate node embeddings $H \in \mathbb{R}^{n \times c}$, where $c$ is the number of labels and the $i$-th row in $H$ indicates the embedding vector $h_i$ for node $v_i$. After that, for the $j$-th label $l_j$, we compute its prototype vector $r_j$ by averaging the embeddings of nodes labeled as $l_j$ in the clean training set $Y_C$. Finally, node embeddings and label prototype vectors are fused to generate $\bar{Y}$.

For $Q_\phi(\bar{Y}|Y_C, G)$, given a node $v_i$, we summarize the process to generate $\bar{y}_i$ as: (1) if $v_i \in \mathcal{T}_C$ & $y_i = l_j$, $\bar{y}_i = \frac{1}{2}(h_i + r_j)$; (2) otherwise, $\bar{y}_i = \frac{1}{2}(h_i + \bar{r}_i)$. Here, $\bar{r}_i = \arg\max_{r_j} h_i^T r_j$, which denotes the most similar label prototype vector to node $v_i$.

Similarly, for $Q_\phi(\bar{Y}|Y_C, G, Y_N)$, we also describe the process to generate $\bar{y}_i$ for node $v_i$ as: (1) if $v_i \in \mathcal{T}_C$ & $y_i = l_j$, $\bar{y}_i = \frac{1}{2}(h_i + r_j)$; (2) if $v_i \in \mathcal{T}_N$ & $y_i = l_j$, $\bar{y}_i = \frac{1}{2}[h_i + \alpha r_j + (1 - \alpha)\bar{r}_i]$; (3) otherwise, $\bar{y}_i = \frac{1}{2}(h_i + \bar{r}_i)$. In particular, $\bar{r}_i$ is used to alleviate the adverse effect of noise labels for nodes in $\mathcal{T}_N$, and $\alpha = cosine(h_i, r_j)$ is employed to control the importance of $r_j$ and $\bar{r}_i$.

Obviously, the more nodes in $\mathcal{T}_C$, the more accurate $r$ will be. Therefore, in each training epoch, we expand $\mathcal{T}_C$ by adding nodes from $\mathcal{T}_N$ with high confidence. Specifically, for each node in $\mathcal{T}_N$, we measure the similarity between its predicted label and given label in $Y_N$. When the similarity is greater than a pre-set threshold $\delta$, we add it to $\mathcal{T}_C$. Additionally, we reset $\mathcal{T}_C$ in each epoch to avoid adding too many nodes with noisy labels.

### 4.4 DECODER

Although we use $P_\varphi(Y_C|\bar{Y}, G)$ and $P_\varphi(Y_C|\bar{Y}, G, Y_N)$ in the ELBOs (see Eqs. 3 and 4), the decoder $P_\varphi$ can generate labels $\hat{Y}$ for all the nodes in the graph. On the one hand, considering $P_\varphi(\cdot|\bar{Y}, G)$, we reconstruct $\hat{y}_i$ for node $v_i$ by $\hat{y}_i = \frac{1}{2}(h_i + \hat{r}_i)$, where $\hat{r}_i = \sum_{j=1}^{c} \bar{y}_{ij} r_j$. Here, we aggregate all prototype vectors $r_j$ with probability $\bar{y}_{ij}$ as weight. On the other hand, for $P_\varphi(\cdot|\bar{Y}, G, Y_N)$, $Y_N$ is given as a known conditional. When reconstructing the label $\hat{y}_i$ for a node $v_i \in \mathcal{T}_N$, we have to consider both the hidden variable $\bar{y}_i$ and the given label $y_i$. When $\bar{y}_i$ and $y_i$ are similar, the given label is more likely to be a clean one; otherwise, not. Therefore, the process to reconstruct $\hat{y}_i$ for node $v_i$ is adjusted as $\hat{y}_i = \frac{1}{2}(h_i + \hat{r}_i)$: (1) if $v_i \in \mathcal{T}_N$, $\hat{r}_i = \sum_{j=1}^{c}(\beta y_{ij} + (1 - \beta)\bar{y}_{ij})r_j$; (2) otherwise, $\hat{r}_i = \sum_{j=1}^{c} \bar{y}_{ij} r_j$. Note that $\beta = cosine(\bar{y}_i, y_i)$ measures the cosine similarity between $\bar{y}_i$ and $y_i$, which aims to assign large (small) weights to clean (noisy) labels.

### 4.5 PRIOR KNOWLEDGE

Different from the vanilla VAE that uses $\mathcal{N}(0, 1)$ as the prior knowledge, in our framework, we instead use $P_\theta(\bar{Y}|G)$ and $P_\theta(\bar{Y}|G, Y_N)$. For the former, based on the input graph $G$, we can run GNNs to get node embeddings $H \in \mathbb{R}^{n \times c}$ and set $\bar{Y} = H$. For the latter, although $Y_N$ contains noise, there still exist many informative clean labels that can be utilized. Specifically, for each label $l_j$, we first compute the corresponding prototype vector $r_j$ by averaging the embeddings of nodes labeled as $l_j$ in $\mathcal{T}_N$. Then we describe the prior knowledge of $\bar{y}_i$ as: (1) if $v_i \in \mathcal{T}_N$ & $y_i = l_j$, $\bar{y}_i = \frac{1}{2}(h_i + r_j)$; (2) otherwise, $\bar{y}_i = h_i$.

---

[2] For simplicity, variance in the Gaussian distribution is assumed to be 0.

# 5 EXPERIMENTS

In this section, we evaluate the performance of PRGNN on 8 benchmark datasets. We compare methods on the node classification task with classification accuracy as the measure. The analysis on the time and space complexity of the model is included in Appendix G.

## 5.1 EXPERIMENTAL SETTINGS

**Datasets.** We evaluate the performance of PRGNN using eight benchmark datasets Sen et al. (2008); Pei et al. (2020); Lim et al. (2021), including homophilous graphs *Cora*, *CiteSeer*, *PubMed*, *ogbn-arxiv*, and heterophilous graphs *Chameleon*, *Actor*, *Squirrel*, *snap-patents*. Here, ogbn-arxiv and snap-patents are large-scale datasets while others are small-scale ones. Details on these graph datasets are shown in Appendix A. For small-scale datesets, we follow Xia et al. (2021a) to split the datasets into 4:4:2 for training, validation and testing, while for large-scale datasets, we use the same training/validation/test splits as provided by the original papers. For fairness, we also conduct experiments on Cora, CiteSeer and PubMed following the standard semi-supervised learning setting, where each class only have 20 labeled nodes.

**Setup.** To show the model robustness, we corrupt the labels of training sets with two types of label noises. **Uniform Noise**: The label of each sample is independently changed to other classes with the same probability $\frac{p}{c-1}$, where $p$ is the noise rate and $c$ is the number of classes. **Flip Noise**: The label of each sample is independently flipped to similar classes with total probability $p$. In our experiments, we randomly select one class as a similar class with equal probability. For small-scale datasets, following Xia et al. (2021a), we only use nearly 25 labeled nodes from the validation set as the clean training set $\mathcal{T}_C$, where each class has the same number of samples. For large-scale datesets, a clean label set of 25 nodes is too small, so we set the size to be $0.2\%$ of the training set size. For fairness, we use the same backbone GCN for PRGNN and other baselines. Due to the space limitation, we move implementation setup to Appendix B.

**Baselines.** We compare PRGNN with multiple baselines using the same network architecture. These baselines are representative, which include **Base models**: GCN Kipf & Welling (2016) and H2GCN Zhu et al. (2020); **Robust loss functions against label noise**: GCE loss Zhang & Sabuncu (2018) and APL Ma et al. (2020); **Typical and effective methods in CV**: Co-teaching plus Yu et al. (2019a); **Methods that handle noisy labels on graphs**: D-GNN Nt et al. (2019), NRGNN Dai et al. (2021) and LPM Xia et al. (2021a). For those baselines that do not consider the clean label set (GCN, GCE loss, APL, Co-teaching plus, D-GNN, NRGNN), we finetune them on the initial clean set after the model has been trained on the noisy training set for a fair comparison.

## 5.2 NODE CLASSIFICATION RESULTS

We perform node classification task, and compare PRGNN-v1 and PRGNN-v2 with other baselines under two types of label noise and four different levels of noise rates to demonstrate the effectiveness of our methods. Table 1 and 2 summarize the performance results on 6 small-scale datasets, from which we observe:

(1) Compared with the base model GCN, GCE and APL generally perform better. This shows the effectiveness of robust loss function. However, as the noise rate increases, their performance drops significantly. For example, with 80% uniform noise on Cora, their accuracy scores are around 0.6, while the best accuracy (PRGNN-v2) is 0.7598.

(2) PRGNN clearly outperforms D-GNN, NRGNN and LPM in heterophilous graphs. For example, with 80% flip noise on Chameleon, the accuracy scores of D-GNN, NRGNN and LPM are 0.3667, 0.3610 and 0.3782, respectively, while the best accuracy score (PRGNN-v2) is 0.4342. This is because they heavily rely on the label smoothness assumption that does not hold in heterophilous graphs.

(3) PRGNN-v2 generally performs better than PRGNN-v1 at high noise rates. For example, with 80% flip noise on Cora and PubMed, the accuracy scores of PRGNN-v1 are 0.6481 and 0.7255, while that of PRGNN-v2 are 0.6731 and 0.7597, respectively. This is because PRGNN-v1 maximizes $P(Y_C|G)$, which generates $Y_C$ based on $G$ only (see Figure 1(a)), and implicitly restricts the closeness between $\bar{Y}$ and $Y_N$ with regularization terms to employ the useful information from $Y_N$.

Table 1: Comparison with baselines in test accuracy (%) with *uniform noise* on *small-scale datasets*. The best and the runner-up results are highlighted in **bold** and underlined respectively.

| Datasets | $p$ | GCN | Coteaching+ | GCE | APL | DGNN | NRGNN | LPM | PRGNN-v1 | PRGNN-v2 |
|---|---|---|---|---|---|---|---|---|---|---|
| Cora | 0.2 | 86.07(0.13) | 83.03(0.19) | 85.10(0.09) | 86.26(0.05) | 87.20(0.97) | 86.42(0.26) | 87.46(0.11) | 87.50(0.67) | **87.53(0.38)** |
| | 0.4 | 82.48(0.14) | 71.68(0.21) | 82.89(0.07) | 82.01(0.13) | 83.69(0.74) | 83.91(1.39) | 83.95(0.15) | 84.63(0.22) | **84.83(1.03)** |
| | 0.6 | 75.88(0.15) | 50.05(0.31) | 76.16(0.15) | 74.49(0.11) | 80.00(1.35) | 80.33(2.06) | 79.66(0.22) | 80.40(1.52) | **81.36(1.37)** |
| | 0.8 | 58.81(0.22) | 36.39(0.44) | 60.43(0.21) | 58.72(0.25) | 72.07(1.81) | 72.77(3.57) | 63.38(0.27) | 75.51(1.95) | **75.98(1.85)** |
| CiteSeer | 0.2 | 76.24(0.07) | 75.49(0.24) | 76.54(0.09) | 74.32(0.17) | 76.19(0.63) | 76.25(0.45) | 77.07(0.06) | **77.23(0.69)** | 77.01(0.43) |
| | 0.4 | 73.42(0.21) | 72.71(0.13) | 74.06(0.18) | 71.77(0.15) | 75.62(1.24) | 74.80(1.43) | 75.19(0.15) | 76.01(1.00) | **76.13(0.58)** |
| | 0.6 | 68.13(0.19) | 66.63(0.41) | 69.18(0.24) | 66.78(0.22) | 72.79(0.98) | 70.69(0.70) | 70.05(0.11) | 72.97(0.85) | **73.28(0.37)** |
| | 0.8 | 56.12(0.28) | 56.27(0.36) | 58.48(0.31) | 56.08(0.34) | 65.20(1.69) | 67.30(1.11) | 61.71(0.22) | **68.52(1.17)** | 67.51(1.56) |
| PubMed | 0.2 | 86.17(0.13) | 85.23(0.21) | 86.11(0.14) | 85.86(0.20) | **86.93(0.20)** | 85.60(0.24) | 86.18(0.15) | 86.63(0.18) | 86.01(0.16) |
| | 0.4 | 85.00(0.26) | 84.30(0.49) | 85.16(0.32) | 85.48(0.24) | 85.70(0.21) | 82.12(1.39) | **86.01(0.24)** | 85.81(0.14) | 85.85(0.21) |
| | 0.6 | 83.95(0.44) | 82.69(0.40) | 83.99(0.67) | 84.63(0.43) | 84.62(0.25) | 80.33(2.06) | 84.17(0.07) | **84.69(0.27)** | 84.32(0.37) |
| | 0.8 | 81.75(0.69) | 80.18(0.13) | 81.89(0.67) | 82.28(0.61) | 82.41(0.68) | 78.83(3.57) | 82.08(0.96) | **82.62(0.42)** | 82.38(0.52) |
| Chameleon | 0.2 | 57.54(1.06) | 56.67(1.43) | 57.54(1.47) | 58.99(1.15) | 56.18(2.26) | 50.70(1.13) | 55.39(2.68) | **60.03(1.68)** | 59.97(0.76) |
| | 0.4 | 55.13(1.68) | 53.90(4.24) | 55.48(2.24) | 56.18(1.45) | 53.77(2.83) | 44.78(1.33) | 50.04(2.93) | **57.88(2.43)** | 57.23(0.99) |
| | 0.6 | 50.35(1.74) | 49.78(1.90) | 50.22(1.74) | 49.96(1.70) | 48.55(1.67) | 40.48(4.66) | 48.20(2.13) | 51.89(2.22) | **52.41(0.97)** |
| | 0.8 | 41.40(1.53) | 41.36(3.66) | 41.23(1.54) | 40.79(2.11) | 41.01(2.96) | 35.22(2.99) | 40.48(2.16) | 44.87(3.40) | **45.61(3.37)** |
| Actor | 0.2 | 31.50(0.45) | 30.29(0.66) | 31.54(0.54) | 29.93(0.31) | 31.46(0.53) | 30.93(0.84) | 28.97(2.09) | **32.25(0.55)** | 32.10(0.29) |
| | 0.4 | **31.13(0.48)** | 30.28(0.50) | 30.88(0.40) | 29.26(0.27) | 30.49(1.29) | 29.09(0.58) | 27.63(2.09) | 30.96(0.42) | 31.08(0.39) |
| | 0.6 | 30.03(0.85) | 29.53(0.59) | 30.05(0.68) | 29.12(0.85) | 29.92(0.86) | 28.36(0.79) | 27.58(1.36) | 31.59(0.77) | **32.84(0.61)** |
| | 0.8 | 28.70(0.78) | 27.92(0.80) | 28.71(0.94) | 28.11(1.08) | 28.07(0.81) | 28.09(0.81) | 26.89(0.32) | 29.51(0.38) | **30.92(0.23)** |
| Squirrel | 0.2 | 36.20(0.64) | 35.93(3.08) | 35.98(0.83) | 32.24(4.39) | 40.81(2.17) | 32.51(0.92) | 32.05(1.36) | 39.48(0.57) | **40.90(0.39)** |
| | 0.4 | 34.31(0.84) | 35.49(1.41) | 34.20(0.59) | 31.99(2.58) | 35.37(1.55) | 31.57(1.91) | 32.12(1.90) | **37.60(0.85)** | 36.48(1.37) |
| | 0.6 | 31.68(1.31) | 32.93(1.37) | 31.70(1.01) | 29.55(0.58) | 32.79(1.45) | 30.49(1.27) | 28.68(2.66) | 33.06(0.83) | **33.28(1.14)** |
| | 0.8 | 29.88(1.62) | 30.82(1.36) | 29.61(1.42) | 28.61(1.01) | 28.70(3.08) | 28.57(1.26) | 26.13(0.83) | 31.68(0.94) | **32.64(1.48)** |

Table 2: Comparison with baselines in test accuracy (%) with *flip noise* on *small-scale datasets*.

| Datasets | $p$ | GCN | Coteaching+ | GCE | APL | DGNN | NRGNN | LPM | PRGNN-v1 | PRGNN-v2 |
|---|---|---|---|---|---|---|---|---|---|---|
| Cora | 0.2 | 82.89(0.14) | 81.37(0.21) | 83.21(0.13) | 82.70(0.79) | 83.99(0.77) | 85.09(0.43) | 86.95(0.12) | 86.87(0.57) | **87.01(0.32)** |
| | 0.4 | 67.39(0.42) | 53.00(0.51) | 67.80(0.37) | 65.00(1.73) | 74.43(3.68) | 71.51(3.26) | 78.97(0.33) | **80.02(1.57)** | 79.34(2.06) |
| | 0.6 | 51.99(1.13) | 48.97(9.05) | 57.82(2.50) | 59.19(4.25) | 61.11(3.04) | 64.58(4.83) | 60.63(3.72) | 69.69(1.46) | **71.55(3.63)** |
| | 0.8 | 41.33(1.12) | 47.56(2.80) | 50.52(3.51) | 50.96(2.58) | 62.88(7.27) | 55.76(8.12) | 42.19(4.56) | 64.81(1.61) | **67.31(3.11)** |
| CiteSeer | 0.2 | 75.08(0.22) | 74.66(0.21) | 76.36(0.20) | 73.38(0.13) | 76.04(0.61) | 76.91(0.18) | 76.39(0.14) | **76.98(0.38)** | 76.24(0.56) |
| | 0.4 | 61.41(0.23) | 60.59(0.33) | 63.66(0.46) | 60.81(0.52) | 65.41(1.99) | 64.95(1.33) | 68.71(0.39) | **71.68(1.06)** | 71.32(0.59) |
| | 0.6 | 36.16(2.17) | 52.01(3.13) | 56.46(2.99) | 58.79(4.94) | 60.15(3.79) | 56.40(6.75) | 60.71(3.55) | 67.80(1.33) | **68.35(1.62)** |
| | 0.8 | 30.54(2.79) | 31.23(7.26) | 41.59(3.54) | 42.97(6.09) | 60.09(4.60) | 47.67(7.12) | 44.07(3.52) | 64.58(2.10) | **66.67(2.15)** |
| PubMed | 0.2 | 85.55(0.24) | 84.61(0.22) | 85.47(0.06) | 85.52(0.06) | **86.66(0.22)** | 81.95(0.09) | 85.58(0.13) | 85.70(0.10) | 85.76(0.21) |
| | 0.4 | 80.88(0.32) | 73.99(0.33) | 80.03(0.42) | 70.08(0.16) | 81.45(1.63) | 75.89(0.39) | 83.15(0.36) | 83.32(0.30) | **83.52(0.40)** |
| | 0.6 | 60.40(2.79) | 58.27(11.61) | 63.55(2.09) | 65.75(3.50) | 74.04(4.55) | 61.31(2.06) | 71.20(4.33) | **77.60(2.73)** | 77.44(3.27) |
| | 0.8 | 51.17(3.85) | 44.22(5.58) | 61.42(3.90) | 63.96(2.98) | 70.52(4.70) | 58.19(3.57) | 59.19(2.62) | 72.55(2.41) | **75.97(2.78)** |
| Chameleon | 0.2 | 58.73(0.09) | 41.01(2.95) | 58.68(0.60) | 54.21(0.16) | 57.32(1.54) | 52.28(0.91) | 55.75(2.06) | 59.34(1.40) | **59.42(1.63)** |
| | 0.4 | 53.51(0.52) | 35.35(2.79) | 53.68(0.51) | 38.42(3.96) | 47.11(2.06) | 45.13(1.17) | 49.47(3.42) | 53.70(2.02) | **53.76(2.83)** |
| | 0.6 | 44.17(2.95) | 27.68(3.56) | 43.99(1.02) | 34.43(2.91) | 39.87(2.44) | 36.84(3.10) | 42.52(2.92) | 44.56(1.62) | **45.35(1.31)** |
| | 0.8 | 37.85(2.10) | 24.30(1.63) | 38.03(3.56) | 35.13(3.02) | 36.67(3.98) | 36.10(3.21) | 37.82(6.41) | 41.10(3.16) | **43.42(2.05)** |
| Actor | 0.2 | 31.37(0.29) | 30.14(0.45) | 31.25(0.12) | 29.66(0.44) | 30.55(0.53) | 27.92(0.29) | 27.00(0.31) | **31.75(1.39)** | 31.30(1.09) |
| | 0.4 | 28.66(1.29) | 27.87(0.64) | 28.83(0.98) | 26.50(0.16) | 27.75(1.05) | 26.42(0.98) | 23.37(1.91) | **29.55(0.81)** | 29.05(0.83) |
| | 0.6 | 26.96(0.46) | 25.74(1.62) | 26.71(0.23) | 26.66(0.77) | 26.89(2.33) | 25.00(1.14) | 25.54(1.28) | 28.54(0.94) | **28.79(0.51)** |
| | 0.8 | 26.67(0.27) | 26.92(0.73) | 26.72(0.26) | 26.55(0.79) | 26.93(2.31) | 23.57(0.97) | 22.47(2.71) | 28.50(0.86) | **28.72(0.43)** |
| Squirrel | 0.2 | 35.97(0.28) | 33.58(0.52) | 35.77(0.43) | 26.01(3.52) | 40.00(0.77) | 33.14(2.14) | 32.28(0.78) | **41.28(0.37)** | 40.73(0.62) |
| | 0.4 | 31.74(0.34) | 30.26(2.66) | 31.51(0.57) | 24.57(1.23) | 34.31(1.76) | 31.35(1.47) | 28.45(1.35) | **37.21(1.25)** | 36.50(1.82) |
| | 0.6 | 28.01(0.80) | 26.86(1.17) | 28.59(0.83) | 24.03(1.33) | 29.97(1.02) | 28.17(1.71) | 25.96(2.30) | 33.05(0.93) | **33.72(1.01)** |
| | 0.8 | 26.32(0.97) | 25.65(2.12) | 25.98(1.53) | 23.90(0.78) | 28.36(2.01) | 26.95(1.00) | 22.44(2.44) | 31.84(0.49) | **32.22(0.82)** |

In this way, when the number of erroneous labels in $Y_N$ is large, $\bar{Y}$ will be noisy and further degrade the performance of generating $Y_C$. However, PRGNN-v2 maximizes $P(Y_C|G, Y_N)$, directly incorporating $Y_N$ in the Bayesian network to generate $Y_C$ (see Figure 1(b)). In particular, when generating $\bar{Y}$, PRGNN-v2 can leverage the power of GNNs to correct noise labels in $Y_N$ and obtain a high-quality $\bar{Y}$, which leads to better reconstruction of $Y_C$.

(4) PRGNN achieves the best or runner-up performance in all 48 cases. This shows that it can consistently provide superior results on datasets in a wide range of diversity. On the one hand, PRGNN disregards the label smoothness assumption for noise correction, which leads to its wide applicability in both homophilous and heterophilous graphs. On the other hand, PRGNN is modeled based on probabilistic graphical model and Bayesian framework, which can model uncertainty and are thus less sensitive to data noise.

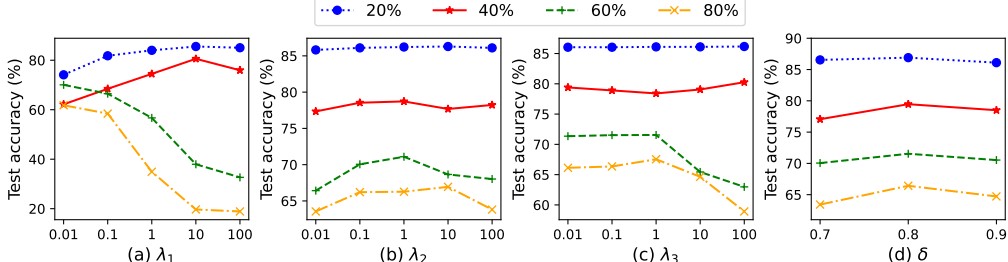

Figure 2: Hyper-parameter sensitivity analysis

For experiments on the large-scale datasets and in the standard semi-supervised learning setting, we observe similar results as above. Therefore, due to the space limitation, we move the corresponding results to Appendix D and E, respectively.

### 5.3    HYPER-PARAMETER SENSITIVITY ANALYSIS

We further perform a sensitivity analysis on the hyper-parameters of our method PRGNN-v2. In particular, we study four key hyper-parameters: the weights for three additional losses besides ELBO, $\lambda_1$, $\lambda_2$, $\lambda_3$, representing the importance of $w\mathcal{L}_{CE}(P_\varphi, Y_N)$, $\mathcal{L}_{CE}(P_\theta, Y_N)$ and $\mathcal{L}_{Cont}(Q_\phi, P_\theta)$ respectively, and the threshold $\delta$ that controls whether nodes can add to $\mathcal{T}_C$. In our experiments, we vary one parameter each time with others fixed. Figure 2 illustrates the results under flip noise ranging from 20% to 80% on Cora. From the figure, we see that

**(1)** As $\lambda_1$ increases, PRGNN-v2 achieves better performance in low noise-rate while achieving worse performance in high noise-rate. This is because there are many incorrect labels in $Y_N$ when the noise rate is high, which may play a great role in misleading the reconstruction of $Y_C$. **(2)** Under high noise rates, PRGNN-v2 performs poorly when $\lambda_2$ is too small ($\lambda_2 = 0.01$) or too large ($\lambda_2 = 100$). This is due to the fact that when $\lambda_2$ is too small, the prior knowledge fails to provide effective positive guidance, while when $\lambda_2$ is too large, the potential erroneous information contained in the prior knowledge can have a detrimental effect and lead to a decrease in performance. **(3)** Although the test accuracy decreases when $\lambda_3$ is set large in high noise rates, PRGNN-v2 can still give stable performances over a wide range of parameter values from [0.01, 1]. **(4)** As $\delta$ increases, the test accuracy first increases and then decreases. This is because when $\delta$ is small, a lot of noise-labeled nodes will be added to $\mathcal{T}_C$, and when $\delta$ is large, more clean-labeled nodes will not be added to $\mathcal{T}_C$, resulting in a large deviation in the prototype vector, which will cause poor performance.

### 5.4    ABLATION STUDY

We conduct an ablation study on PRGNN-v2 to understand the characteristics of its main components. One variant does not consider the useful information from $Y_N$, training the model without $\mathcal{L}_{CE}(P_\varphi, Y_N)$. We call this variant **PRGNN-nl** (**n**o $\mathcal{L}_{CE}(P_\varphi, Y_N)$). Another variant training the model without $\mathcal{L}_{CE}(P_\theta, Y_C)$, which helps us understand the importance of introducing prior knowledge. We call this variant **PRGNN-np** (**n**o **p**rior knowledge). To show the importance of the contrastive loss, we train the model without $\mathcal{L}_{Cont}(P_\theta, Y_C)$ and call this variant **PRGNN-nc** (**n**o **c**ontrastive loss). Moreover, we consider a variant of PRGNN-v2 that applies $\mathcal{L}_{CE}(P_\theta, Y_N)$ directly, without weight $w$. We call this variant **PRGNN-nw** (**n**o **w**eight). Finally, we ignore the problem of semantic space inconsistency between variables, directly concatenating features with one-hot encoded labels as inputs of GNNs. This variant helps us evaluate the effectiveness of introducing label prototype vectors to generate $\bar{Y}$ and $Y_C$. We call this variant **PRGNN-nv** (**n**o prototype **v**ectors). We compare PRGNN-v2 with these variants in the 80% noise rate on all datasets. The results are given in Figure 3. From it, we observe:

**(1)** PRGNN-v2 beats PRGNN-nl in all cases. This is because $Y_N$ contains a portion of clean labels, which can well guide the reconstruction of $Y_C$. **(2)** PRGNN-v2 achieves better performance than PRGNN-np. This further shows the importance of using $Y_C$ to guide prior knowledge. **(3)** PRGNN-v2 performs better than PRGNN-nc. This shows that the contrastive loss can leverage the

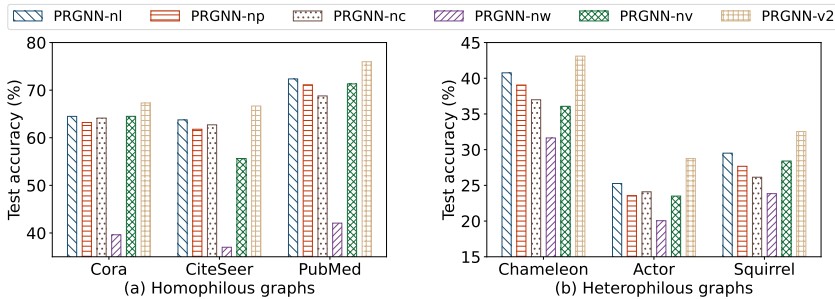

Figure 3: The ablation study results on six datasets with 80% flip noise.

extra supervision information from massive unlabeled data and maximize the agreement of node predictions generated from $Q_\phi(\bar{Y}|Y_C, G, Y_N)$ and $P_\theta(\bar{Y}|G, Y_N)$. **(4)** PRGNN-v2 clearly outperforms PRGNN-nw in all datasets. PRGNN-nw, which ignores the noisy labels in $Y_N$, directly applies cross-entropy loss between the reconstructed labels and $Y_N$. Since there are many incorrect labels in $Y_N$, it will negatively affect the reconstructed labels. **(5)** PRGNN-v2 outperforms PRGNN-nv. This shows the effectiveness of mapping node features and labels into the same low-dimensional space instead of directly concatenating features with one-hot encoded labels as inputs of GNNs.

## 5.5 STUDY ON THE SIZE OF THE CLEAN LABEL SET

We next study the sensitivity of PRGNN on the size of the clean label set. As can be seen in Figure 4(a), our method can achieve very stable performance over a wide range of set sizes on both CiteSeer (homophilous graph) and Chameleon (heterophilous graph) in various noise rates. Given only 20 clean labels, PRGNN can perform very well. With the increase of the clean set size, there only brings marginal improvement on the test accuracy. This further shows that the problem scenario we set is meaningful and feasible. We only need to obtain an additional small set of clean labels at low cost to achieve superior results. We also evaluate the robustness of PRGNN against other baselines w.r.t. the clean label set size. Figure 4(b) shows the results with 80% flip noise. From the figure, we observe that PRGNN consistently outperforms other baselines in all cases, which further verifies the robustness of PRGNN.

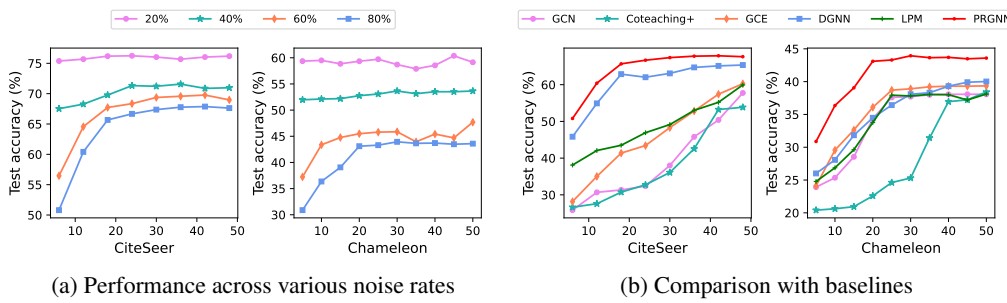

(a) Performance across various noise rates      (b) Comparison with baselines

Figure 4: Robustness study of PRGNN w.r.t. the clean label set size

## 6 CONCLUSION

In this paper, we proposed PRGNN, which is the first probabilistic graphical model based framework for robust GNNs against noisy labels. It disregards the label smoothness assumption and can be applied in both graphs with homophily and heterophily. We first maximized $P(Y_C|G)$ and employed $Y_N$ in regularization terms only. To further leverage clean labels in $Y_N$, we incorporated $Y_N$ in the Bayesian network to generate $Y_C$ and maximized $P(Y_C|G, Y_N)$. We also used label prototype vectors to ensure that labels and nodes are embedded into the same low-dimensional space. Finally, we conducted extensive experiments to show that PRGNN achieves robust performance under different noise types and rates on various datasets.

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

## A  DATASETS

We summarize the statistics of the datasets used in experiments in Table 3.

Table 3: Statistics of the datasets.

| Datasets | Cora | Citeseer | Pubmed | ogbn-arxiv | Chameleon | Actor | Squirrel | snap-patents |
|---|---|---|---|---|---|---|---|---|
| #Nodes | 2,708 | 3,327 | 19,717 | 169,343 | 2,277 | 7,600 | 5,201 | 2,923,922 |
| #Edges | 5,278 | 4,552 | 44,324 | 1,166,243 | 31,421 | 26,752 | 198,493 | 13,975,788 |
| #Features | 1433 | 3703 | 500 | 128 | 2325 | 931 | 2089 | 269 |
| #Classes | 7 | 6 | 3 | 40 | 5 | 5 | 5 | 5 |

## B  IMPLEMENTATION DETAILS

We implement PRGNN with PyTorch and adopt the Adam optimizer to train the model. We perform a grid search to fine-tune hyper-parameters based on the results on the validation set. The search space of these hyper-parameters is listed in Table 4. Further, for other competitors (GCN, GCE, APL, Coteaching, LPM), some of their results are directly reported from Xia et al. (2021a) (Cora, CiteSeer with uniform noise ranging from 20% to 80% and Cora, CiteSeer, PubMed with flip noise ranging from 20% to 40%). For other cases, we fine-tune the model hyper-parameters with the codes released by their original authors. For fair comparison, we report the average results with standard deviations of 5 runs for all experiments. We run all the experiments on a server with 32G memory and a single Tesla V100 GPU.

Table 4: Grid search space.

| Notation | Range |
|---|---|
| learning_rate | {1e-5, 1e-4, 1e-3, 1e-2} |
| weight_decay | {5e-5, 5e-4, 5e-3, 5e-2} |
| hidden_number | {32, 64, 128, 256} |
| dropout | [0.2, 0.8] |
| $\lambda_1$ | {0.1, 0.5, 1, 1.5, 5, 10, 100} |
| $\lambda_2$ | {0.1, 0.5, 1, 1.5, 5, 10, 100} |
| $\lambda_3$ | {0.1, 0.5, 1, 1.5, 5, 10, 100} |
| $\delta$ | {0.7, 0.8, 0.9} |

## C  DETAILS OF THE CONTRASTIVE LOSS

The contrastive loss $\mathcal{L}_{Cont}$ is utilized to leverage the extra supervision information from massive unlabeled data. Specifically, taking PRGNN-v1 as an example, we maximize the agreement of predictions of the same node that are generated from $Q_\phi(\bar{Y}|Y_C, G)$ and $P_\theta(\bar{Y}|G)$. For notation simplicity, we denote the predictions as $y_i$ and $\tilde{y}_i$ for each node $v_i$, respectively. Meanwhile, we pull the predictions of different node pairs away. As a result, the pairwise contrastive loss between $y_i$ and $\tilde{y}_i$ can be defined as

$$\mathcal{L}_{PC}(y_i, \tilde{y}_i) = -log \frac{exp(\langle y_i, \tilde{y}_i \rangle /\tau)}{exp(\langle y_i, \tilde{y}_i \rangle /\tau) + \sum_{j=1}^{n} \mathbb{1}_{[j \neq i]} exp(\langle y_i, \tilde{y}_j \rangle /\tau) + \sum_{j=1}^{n} \mathbb{1}_{[j \neq i]} exp(\langle y_i, y_j \rangle /\tau)} \tag{7}$$

where $\langle \cdot, \cdot \rangle$ denotes the inner product and $\tau$ is a temperature parameter. $\tau$ for PRGNN is set to be 0.5. Based on Equation 7, the overall contrastive objective to be minimized is

$$\mathcal{L}_{Cont} = \frac{1}{2n} \sum_{i=1}^{n} (\mathcal{L}_{PC}(y_i, \tilde{y}_i) + \mathcal{L}_{PC}(\tilde{y}_i, y_i)) \tag{8}$$

Table 5: Comparison with baselines in test accuracy (%) with *flip noise* on *large-scale datasets*.

| Datasets | $p$ | GCN | Coteaching+ | GCE | APL | DGNN | NRGNN | LPM | PRGNN-v1 | PRGNN-v2 |
|---|---|---|---|---|---|---|---|---|---|---|
| ogbn-arxiv | 0.2 | 54.73(0.28) | 52.71(0.35) | 55.22(0.29) | 55.68(0.37) | 53.49(0.48) | OOM | 53.61(0.78) | **56.40(0.39)** | 55.85(0.26) |
| | 0.4 | 53.49(0.91) | 50.12(0.83) | 54.03(1.38) | 53.79(0.62) | 49.97(1.36) | OOM | 48.83(0.82) | **55.67(1.08)** | 54.82(0.85) |
| | 0.6 | 35.01(1.03) | 35.27(0.83) | 36.71(0.79) | 37.26(0.84) | 32.90(0.95) | OOM | 34.16(1.12) | 41.75(1.14) | **44.28(0.94)** |
| | 0.8 | 33.81(0.95) | 31.65(1.27) | 31.20(0.84) | 35.27(0.81) | 29.96(1.83) | OOM | 35.63(0.84) | 40.11(0.99) | **42.39(0.74)** |
| snap-patents | 0.2 | 42.24(0.11) | 41.55(0.18) | 42.40(0.27) | 42.36(0.19) | 43.08(0.15) | OOM | 39.62(0.32) | **43.26(0.12)** | 43.05(0.11) |
| | 0.4 | 36.23(0.29) | 36.57(0.31) | 36.29(0.25) | 37.88(0.30) | 37.06(0.62) | OOM | 31.29(0.74) | 39.32(0.26) | **39.70(0.33)** |
| | 0.6 | 33.21(1.49) | 31.55(1.98) | 34.28(0.94) | 34.05(1.27) | 25.19(2.87) | OOM | 28.11(3.65) | 37.07(0.34) | **38.01(0.81)** |
| | 0.8 | 32.57(1.74) | 30.34(1.22) | 31.90(0.78) | 32.81(1.27) | 23.25(2.99) | OOM | 25.98(1.76) | 36.75(1.53) | **37.82(0.91)** |

## D    EXPERIMENTS ON LARGE-SCALE DATASETS

Table 5 summarizes the classification results on large-scale datasets. From the figure, we see that PRGNN consistently outperforms other baselines on both ogbn-arxiv (homophilous graph) and snap-patents (heterophilous graph). Further, when noise rates are high, the performance gap between PRGNN and baselines gets larger. This shows that PRGNN is more robust against label noise. Further, since the edge prediction module in NRGNN has a time complexity of $O(n^2)$, it fails to run on large-scale datasets due to the out-of-memory (OOM) error.

## E    EXPERIMENTS IN THE STANDARD SEMI-SUPERVISED LEARNING SETTING

To further demonstrate the effectiveness of our methods, we perform node classification task, and compare PRGNN-v1 and PRGNN-v2 with other baselines in the standard semi-supervised learning setting where each class only have 20 labeled nodes for Cora, CiteSeer and PubMed. Table 6 summarizes the performance results, from which we observe that PRGNN clearly outperforms other baselines in all cases.

Table 6: Comparison with baselines in test accuracy (%) with *flip noise* and *standard semi-supervised learning setting*.

| Datasets | $p$ | Coteaching+ | DGNN | NRGNN | LPM | PRGNN-v1 | PRGNN-v2 |
|---|---|---|---|---|---|---|---|
| Cora | 0.2 | 72.55 | 78.65 | 78.14 | 77.27 | **80.30** | 79.23 |
| | 0.4 | 64.56 | 68.63 | 69.75 | 67.82 | **75.12** | 72.68 |
| | 0.6 | 55.12 | 63.38 | 62.52 | 60.93 | 67.94 | **69.00** |
| | 0.8 | 51.28 | 58.35 | 57.51 | 53.08 | 63.06 | **67.57** |
| CiteSeer | 0.2 | 58.43 | 62.42 | 64.36 | 63.29 | 65.43 | **66.63** |
| | 0.4 | 56.06 | 57.37 | 60.23 | 57.81 | 61.86 | **64.45** |
| | 0.6 | 50.18 | 53.15 | 54.35 | 52.03 | 57.01 | **61.31** |
| | 0.8 | 49.80 | 50.31 | 51.34 | 48.92 | 52.25 | **59.58** |
| PubMed | 0.2 | 72.26 | 75.32 | 74.93 | 75.02 | **76.84** | 76.05 |
| | 0.4 | 70.84 | 74.10 | 71.28 | 71.62 | **74.33** | 73.97 |
| | 0.6 | 69.38 | 71.92 | 70.37 | 70.18 | 73.17 | **73.23** |
| | 0.8 | 66.41 | 68.09 | 67.32 | 66.35 | 71.34 | **72.69** |

## F    PRGNN WITH H2GCN AS THE BACKBONE MODEL

Since our framework is flexible to use various GNNs as the backbone model, we also use two-layer H2GCN Zhu et al. (2020) for heterophilous graphs. Table 7 summarizes the performance results with flip noise. For fairness, we replace the corresponding backbones for other models except NRGNN and LPM, which are based on the assumption of label smoothness and can only use GCN as backbone. The results demonstrate that the performance of our proposed method PRGNN still leads other competitors.

## G    TIME AND SPACE COMPLEXITY ANALYSIS

The major time complexity in the PRGNN comes from GNNs and the contrastive loss. Suppose we use one-layer GCN as the backbone. Since the adjacency matrix is generally sparse, let $d_A$ be the

Table 7: Comparison with baselines in test accuracy (%) on heterophilous graphs with *flip noise* and *H2GCN as the backbone GNN*.

| Datasets | $p$ | H2GCN | Coteaching+ | GCE | APL | DGNN | NRGNN | LPM | PRGNN-v1 | PRGNN-v2 |
|---|---|---|---|---|---|---|---|---|---|---|
| **Chameleon** | 0.2 | 55.44(0.90) | 58.20(1.58) | 56.75(1.19) | 58.60(0.85) | 51.54(1.49) | 52.28(0.91) | 55.75(2.06) | **59.69(0.63)** | 59.34(0.61) |
| | 0.4 | 50.04(2.03) | 50.13(0.51) | 51.23(2.48) | 50.66(1.61) | 44.17(3.97) | 45.13(1.17) | 49.47(3.42) | **53.73(1.55)** | 52.76(1.62) |
| | 0.6 | 42.48(3.16) | 37.59(3.53) | 42.84(2.45) | 37.06(3.97) | 34.61(2.24) | 36.84(3.10) | 42.52(2.92) | **44.08(2.33)** | 43.05(2.47) |
| | 0.8 | 37.95(2.58) | 35.66(2.60) | 38.01(3.15) | 33.51(3.76) | 32.24(1.57) | 36.10(3.21) | 37.82(6.41) | **39.12(2.47)** | 38.10(2.35) |
| **Actor** | 0.2 | 35.49(0.45) | 35.78(0.94) | 34.54(1.27) | 33.78(0.76) | 32.22(1.90) | 27.92(0.29) | 27.00(0.31) | 35.82(1.48) | **36.00(0.23)** |
| | 0.4 | 27.70(1.21) | 32.01(0.93) | 28.97(1.13) | 28.34(1.31) | 27.33(1.34) | 26.42(0.98) | 23.37(1.91) | 32.79(1.04) | **33.32(0.23)** |
| | 0.6 | 26.87(1.30) | 29.28(1.87) | 27.33(1.61) | 26.09(0.44) | 26.26(0.24) | 25.00(1.14) | 25.54(1.28) | 31.17(1.62) | **31.54(0.72)** |
| | 0.8 | 26.83(1.30) | 27.84(1.32) | 27.18(1.20) | 24.51(2.82) | 24.92(1.77) | 23.57(0.97) | 22.47(2.71) | 29.93(2.87) | **30.75(1.32)** |
| **Squirrel** | 0.2 | 43.30(0.91) | 43.42(1.24) | 42.44(1.04) | 43.27(0.62) | 39.21(0.61) | 33.14(2.14) | 32.28(0.78) | 43.75(1.20) | **43.90(1.29)** |
| | 0.4 | 36.10(1.01) | 36.18(1.33) | 34.66(1.59) | 32.93(1.52) | 28.93(2.04) | 31.35(1.47) | 28.45(1.35) | **36.75(1.23)** | 36.41(0.99) |
| | 0.6 | 32.33(3.37) | 31.10(2.25) | 32.22(3.17) | 29.07(3.34) | 25.40(2.11) | 28.17(1.71) | 25.96(2.30) | 32.89(2.23) | **33.09(2.19)** |
| | 0.8 | 27.30(4.12) | 27.97(2.17) | 26.34(3.91) | 28.51(2.03) | 25.21(0.74) | 26.95(1.00) | 22.44(2.44) | 30.11(1.87) | **31.54(0.61)** |

average number of non-zero entries in each row of the adjacency matrix. Let $n$ be the number of nodes, $l$ be the raw features dimension, and $c$ be the output layer dimension ($c$ is the number of class labels). Further, let $k$ be the number of selected negative samples. Then, the time complexities for GCN and the contrastive loss are $O(nd_A l + nlc)$ and $O(nkc)$, respectively, which are both linear to the number of nodes $n$.

For the space complexity, we need to store $\bar{Y}$ whose size is $nc$ ($c$ is the number of class labels) and the parameters in GCN. Note that we use three independent GNNs in our framework. We still take one-layer GCN for illustration. For the convolutional layer in each GCN, there exists a learnable transformation matrix of size $lc$. Therefore, the overall space complexity is $O(nc + 3lc)$, which is also widely applicable.

We further conduct experiments to study model efficiency for different methods on two large-scale datasets. Note that we choose methods specially designed for graphs against label noise for fair comparison and they all use GCN as the backbone model. We also take GCN as a reference model. We compare the convergence time (second) in training under flip noise rate = 0.8. We run all the experiments on one V100 GPU. Our results are shown in Table 8. From the table, we see that DGNN is very efficient. This is because based on GCN, DGNN only slightly adjusts the training objective by introducing a learnable noise correction matrix to the cross-entropy function. However, DGNN performs poorly under high noise rates. Further, our proposed methods are more efficient than NRGNN and LPM, which can also achieve the best classification performance (See Table 5). All these results show the superiority of our proposed methods.

Table 8: Comparison with baselines in convergence time (s) on large-scale datasets.

| Datasets | GCN | DGNN | NRGNN | LPM | PRGNN-v1 | PRGNN-v2 |
|---|---|---|---|---|---|---|
| ogbn-arxiv | 14.57 | 17.80 | OOM | 118.45 | 62.05 | 60.93 |
| snap-patents | 124.95 | 133.61 | OOM | 805.05 | 614.85 | 625.04 |

## H  BROADER IMPACTS AND LIMITATIONS

**Broader impacts.** In this paper, we propose a novel probabilistic graphical model based framework PRGNN. It can effectively solve the problem of noisy labels for GNNs, but at the same time, there are also some potential risks involved: 1) Widespread adoption of this model will reduce the need for high-quality markers on crowdsourcing platforms, which could lead to increasing unemployment. 2) Further, lower requirements for labeling quality may result in reduced monitoring and validation of data quality, thereby disrupting the management of the data labeling industry.

**Limitations.** Although our methods solve the problem of noisy labels in both graphs with ho-mophily and heterophily well, implementing them with three independent GNNs could be a draw-back. How to implement our methods with less GNNs is a promising future research topic. Besides, experiments on datasets with real-world label noise are also needed to further show the effectivess of PRGNN.

