# OpenReview forum: "Probabilistic Graphical Model for Robust Graph Neural Networks against Noisy Labels"
_ICLR.cc/2024/Conference — ICLR 2024 Conference Withdrawn Submission_

### Official Review · Reviewer_JeRR · 2023-10-28

**Soundness:** 2 fair
**Presentation:** 3 good
**Contribution:** 2 fair
**Rating:** 5
**Confidence:** 3

**Summary:**

This paper propose a probabilistic graphical model based framework PRGNN for graph neural networks with label noise. Authors disregard the label smoothness assumption for noise correction, which leads to the wide applicability of PRGNN in both homophilous and heterophilous graphs. Extensive experiments on various benchmark datasets validate the effectiveness of the proposed PRGNN.

**Strengths:**

1. This paper studies an important problem.
2. The presentation is good.

**Weaknesses:**

1. The experiment is not sufficient. More baselines published in 2022-2023 should be included, e.g., [1,2].
2. More ablation studies in different noise rates should be included. Now the effectiveness of each compotent is not clear.
3. The motivation of using a probabilistic graphical model for this problem is not clear. What's the key to bring in performance increasement? The proposed methods seems to combine a lot existing things, prototypes, contrastive learning, weight learning.


[1] Learning on Graphs under Label Noise, ICASSP 23.
[2] Robust Training of Graph Neural Networks via Noise Governance, WSDM 23.

**Questions:**

See weakness.

---

### Official Review · Reviewer_ncr4 · 2023-10-31

**Soundness:** 2 fair
**Presentation:** 2 fair
**Contribution:** 2 fair
**Rating:** 3
**Confidence:** 4

**Summary:**

The authors have proposed two methods, PRGNN and PRGNN-v2, for handling noisy labels in graph data. A generative approach is proposed, which is used to predict labels for unlabeled nodes based on variational inference. Experiments on different datasets demonstrate the robustness of the proposed method.

**Strengths:**

- **Attempt at Addressing a Less Explored Area in GNNs.** The attempt to tackle label noise specifically in the context of GNNs, rather than more common settings like images in computer vision, is a direction that enriches the diversity of research in the field.
-  **Introduction of Probabilistic Graphical Models to GNNs.** The conceptual idea of integrating probabilistic graphical models with GNNs for handling label noise remains an interesting proposition. This concept could inspire further research that might rectify the shortcomings of this initial attempt.

**Weaknesses:**

- **Comparison with CausalNL Method.** The paper presents a methodology that appears conceptually similar to CausalNL (presented at NeurIPS 21), which uses a generative process to correct noisy labels. However, while CausalNL deals with image data, this paper focuses on graph data. The key concern here is that, aside from the data type, the proposed method may not sufficiently distinguish itself from CausalNL in terms of the underlying principles and approach. A more explicit and thorough comparison is needed to highlight the novelty and advancements of the proposed method over existing solutions like CausalNL.

- **Counterintuitive Generative Process.**  The generative process, as depicted in Figures 1(a) and 1(b), posits that noisy labels (\(Y_N\)) are causes of clean labels (\(Y_C\)), which is counterintuitive and lacks practical examples or theoretical justification. This assumption contradicts conventional understanding where clean labels typically precede and are corrupted into noisy labels, not the other way around. The paper should offer a clear rationale or model explaining how and why noisy labels can causally lead to clean labels in this context.

- **Lack of Clarity in Generative vs. Inference Processes.** The paper seems to mislabel certain processes, like referring to the encoder as part of the generative process when it actually pertains to inference. This confusion undermines the clarity of the methodological framework. The authors need to distinguish between the generative and inference components of their model more clearly to avoid contradictory statements and ensure the coherence of their proposed method.

- **Insufficient Self-Containment and Explanation of Methodology.**  The paper lacks self-contained explanations, especially in deriving important elements like the Evidence Lower Bound (ELBO). These critical steps are relegated to the appendix, making it difficult for readers to follow the methodology's theoretical foundation. Additionally, the application of techniques like co-teaching+ to graph data is not explained, leaving gaps in the readers' understanding of how the methodology is adapted for the graph context.

- **Inadequate Experimental Comparison and Marginal Improvements.** The experimental section does not provide a robust comparative analysis. Specifically, 5 out of 6 baselines do not use a clean label set, which might not be an apples-to-apples comparison. Additionally, the improvements shown by PRGNN-v1 and PRGNN-v2 are marginal and similar to the methods that don't use clean labels. This raises questions about the practical effectiveness and the added value of the proposed approach.

- **Limited Discussion on Theoretical Underpinnings.** The paper does not delve deeply into the theoretical aspects that underlie the proposed framework. A more rigorous discussion on the probabilistic and Bayesian foundations specific to graph neural networks and how they effectively address noisy labels is needed. This would provide a stronger theoretical base for the paper and justify its approach more convincingly.


_______________
Post Discussion Suggestion:

Dear authors,

I believe that the two weaknesses, Counterintuitive Generative Process and Clarity in Generative vs. Inference Processes, I mentioned above are very important. Please kindly consider addressing these two in the future version, which I believe will greatly improve your paper.
________________

**Questions:**

- Could the authors please elaborate on how PRGNN distinctly advances beyond the methodology used in CausalNL (NeurIPS 21), especially considering that both approaches utilize generative processes for correcting noisy labels? A comparative analysis highlighting specific differences and innovations would be highly beneficial.
- Regarding the assumption that noisy labels $Y_N$ cause clean labels $Y_C$, could the authors provide theoretical justification or practical examples to support this approach? It would be helpful to understand the rationale behind this unconventional model.
- The encoder is referred to as part of the generative process. Could the authors clarify this aspect, as it typically aligns more with inference processes? A clear differentiation between the generative and inference components in your model would greatly enhance the clarity of your proposed method.
- Would it be possible for the authors to include more detailed explanations within the main paper or appendix, particularly regarding the derivation of the Evidence Lower Bound (ELBO) and the adaptation of techniques like co-teaching+ for graph data? This would greatly aid in understanding the theoretical and practical aspects of your approach.
- Could the authors expand on the theoretical underpinnings of their framework, especially why modeling the generative process can help learn clean labels?

---

### Official Review · Reviewer_XrQv · 2023-11-02

**Soundness:** 3 good
**Presentation:** 2 fair
**Contribution:** 2 fair
**Rating:** 5
**Confidence:** 4

**Summary:**

This paper focuses on the problem of dealing with label noises in graph neural network training. In this paper, a PGM-based framework called PRGNN is proposed. Specifically, the proposed PRGNN framework aims to maximize the likelihood of clean labels given a set of noisy labels and a separate set of clean labels. It is argued that the label smoothness assumption for noise correction is not required in the proposed framework, which makes it applicable to both homophilous and heterophilous graphs. Experiments are compared with various baselines on graphs with different homophily levels

**Strengths:**

1. This paper focuses a problem of graph neural network learning with noisy labels. And the authors try to address the challenge caused by the heterophily and investigate methods do not rely on homophily assumption, which is an important open problem.
2. The literature review is comprehensive. The organization of the paper is clear.
3. Extensive experiments are conducted for comparisons and evaluation.

**Weaknesses:**

1. The proposed framework seems to unreasonably complex to me. The motivation of using PGM and Bayesian framework is not clear and convinced.
2. To address the problem of graph heterophily, the proposed solution is manually change suitable backbones. This is a great application of existing heterophilious GNNs. However, the technique contribution in addressing graph heterophily with label noises is marginal.
3. The improvements that the complex proposed method can bring seem marginal. For example, NRGNN often achieve comparable results with the PRGNN on homophilious graphs

**Questions:**

Please refer to the weakness.

---

### Official Review · Reviewer_b48A · 2023-11-05

**Soundness:** 2 fair
**Presentation:** 2 fair
**Contribution:** 3 good
**Rating:** 5
**Confidence:** 3

**Summary:**

The paper investigates the crucial but relatively less explored area of robust graph neural networks (GNNs) with a focus on handling label noise. While extensive research has gone into perturbations and attacks on graph structures, the treatment of label noise remains underrepresented in the literature. One noticeable limitation in existing methods is their heavy reliance on the label smoothness assumption, which often hampers their effectiveness when dealing with diverse and heterogeneous graph structures. Furthermore, their performance tends to degrade significantly under high noise-rate scenarios. This paper presents a novel solution, the Probabilistic Graphical Model-based framework, PRGNN, to address these challenges. The core objective of PRGNN is to maximize the likelihood of labels in a clean label set given a noisy label set. The paper introduces two variants of PRGNN, PRGNN-v1 and PRGNN-v2, each with a unique approach to generating clean labels. PRGNN-v1 solely utilizes the information from the Bayesian network and graphs, while PRGNN-v2 takes a more comprehensive approach by incorporating the noisy label set into the Bayesian network, resulting in a refined clean label generation process. Subsequently, these generative models can be leveraged to predict labels for unlabeled nodes. The authors demonstrate the efficacy of PRGNN through a series of extensive experiments. They systematically evaluate PRGNN's performance under various noise types and rates, as well as across graphs with varying degrees of heterophily. Notably, the paper highlights PRGNN's exceptional performance in high noise-rate scenarios, which is a significant contribution to the field.

**Strengths:**

Strengths:

1. Overall, the paper presents a novel and promising approach to handling label noise in graph neural networks based on probabilistic graphical model.

2. The sequentially introduced models PRGNN-V1 and PRGNN-V2 are clearly depicted and the proposed objectives are well motivated.

3. Experiments are conducted on both noisy Homophilous and Heterophilous graphs. Clear improvements are observed compared to the implemented methods.

**Weaknesses:**

1. Several crucial points remain unclear. First, it is still hard to understand why we need the uncertainty modeling. One might directly apply the encoder-decoder models equipped with the corresponding losses to handle the noisy labels. To be specific, what happens if we only use Q and P_\varphi without the prior distribution, and conduct the training without the prior-related losses? The second question is related. From the experiments in Figure 3, the weighted cross-entropy loss wL_CE(P_\varphi, Y_N) is much more important than others. So, it seems it is much more important to balance the loss between reconstruction of clean labels and that of noisy labels, other than the probabilistic modelling particularly given the label space (which is the random variable we focus on) is of small dimension. Third, the authors claim that the proposed probabilistic graphical model is able to avoid the issue of label smoothness in existing methods. However, it seems this is attributed to the application of prototype label vectors, instead of simple concatenation of labels with node embeddings for message passing. This is also observed in Figure 3. Overall, to make the proposed models effective, the authors introduce various techniques. However, it appears that the one which demonstrates consistent success is not aligned with the contributions claimed by the authors.

2. About the introduction of the baselines, it is unknown which one applies label smoothness, and which one is proposed for Heterophilous graphs.

3. It seems this paper is only applicable for the transductive setting not inductive setting.

4. The last question is on the general research domain for learning from noisy graphs. While it is true that many data we process in practice definitely contain different kinds of noise, the proposed methods are always evaluated in the scenarios where we add artificial perturbation to the clean data. My point is we actually need some real tasks and perhaps some fixed benchmarks to evaluate the performance of different methods, such that we can clearly justify how the progress is made and how the challenges are addressed.

**Questions:**

See the weaknesses above.